# Some Supervision Required:
# Incorporating Oracle Policies in Reinforcement Learning via Epistemic Uncertainty Metrics

## Abstract

An inherent problem of reinforcement learning is performing exploration of an environment through random actions, of which a large portion can be unproductive. Instead, exploration can be improved by initializing the learning policy with an existing (previously learned or hard-coded) oracle policy, offline data, or demonstrations. In the case of using an oracle policy, it can be unclear how best to incorporate the oracle policy's experience into the learning policy in a way that maximizes learning sample efficiency. In this paper, we propose a method termed *Critic Confidence Guided Exploration* (CCGE) for incorporating such an oracle policy into standard actor-critic reinforcement learning algorithms. More specifically, CCGE takes in the oracle policy's actions as suggestions and incorporates this information into the learning scheme when uncertainty is high, while ignoring it when the uncertainty is low. CCGE is agnostic to methods of estimating uncertainty, and we show that it is equally effective with two different techniques. Empirically, we evaluate the effect of CCGE on various benchmark reinforcement learning tasks, and show that this idea can lead to improved sample efficiency and final performance. Furthermore, when evaluated on sparse reward environments, CCGE is able to perform competitively against adjacent algorithms that also leverage an oracle policy. Our experiments show that it is possible to utilize uncertainty as a heuristic to guide exploration using an oracle in reinforcement learning. We expect that this will inspire more research in this direction, where various heuristics are used to determine the direction of guidance provided to learning.

## 1 Introduction

Reinforcement Learning (RL) seeks to learn a policy that maximizes the expected discounted future rewards for Markov Decision Processes (MDPs) (Sutton & Barto, 2018). Unlike supervised learning, which learns a function to map data to labels, RL involves an agent interacting with an environment to learn how to make decisions that optimize a reward signal. While RL has shown great promise in a wide range of applications, it can be challenging to explore complex environments to find optimal policies. Most popular RL methods employ stochastic actions to explore the environment, which can be time-consuming and lead to suboptimal solutions if the agent gets stuck in local minima, which can be true due to the curse of dimensionality or in complex environments where the reward signal is sparse.

One technique to circumvent this issue is by incorporating prior knowledge into the learning process via the use of an oracle policy — a policy that takes better-than-random actions when given a state observation. Such an oracle policy can be obtained from demonstration data through behavioral cloning (Bain & Sammut, 1995), pretrained using a different RL algorithm, or simply hard-coded. That said, it may not be clear how best to incorporate this oracle policy into the learning policy — and more crucially, when to wean the learning policy off the oracle policy. In this paper, we propose a new approach for RL called Critic Confidence Guided Exploration (CCGE) that seeks to address these challenges by using uncertainty to decide when to use the oracle policy to guide exploration versus aiming to simply optimize a reward signal. To our knowledge, this idea of using a heuristic to determine when to utilize an oracle in learning is a first of its kind.

The proposed CCGE algorithm builds upon the popular actor critic framework for RL, where an actor learns a policy that interacts with the environment while the critic aims to learn the value function (Grondman et al., 2012). Many of the most widely used methods in deep RL use the actor critic framework. For instance, they comprise four out of twelve algorithms in OpenAI Baselines Dhariwal et al. (2017) and four out of seven algorithms in Stable-Baselines3 (Raffin et al., 2019). CCGE works by using the critic's prediction error or variance as a proxy for uncertainty. When uncertainty is high, the actor learns from the oracle policy to guide exploration, and when uncertainty is low, the actor aims to maximize the learned value function. Our intuition is that it is more beneficial to allow the learning policy to decide when to follow the oracle policy, as opposed to competing approaches that dictate when this switching happens.

We compare the proposed CCGE algorithm with existing exploration strategies on several benchmark RL tasks for robotics, and our experiments demonstrate that it can lead to better performance and faster convergence in some challenging environments. Specifically, for dense reward environments, we show that CCGE can lead to policies that escape local minima faster through exploration via an oracle policy. We further show that this approach also works for sparse reward environments, being competitive with other similar approaches. Our results show that CCGE has the potential to enable more efficient and effective RL algorithms.

## 2 Related Work

### 2.1 Imitation Learning / Offline Reinforcement Learning Initialized Policies

One method to improve sample efficiency of RL is to initialize training with policies obtained from imitation learning (Silver et al., 2016; Kim et al., 2022; Chang et al., 2021; Kidambi et al., 2021). The goal here is to train a policy via supervised learning on a dataset of states and actions obtained from an optimal oracle policy. This policy can then be used in an environment during finetuning to produce a more optimal policy (Gupta et al., 2019; Lavington et al., 2022). This method can work reasonably well for policy gradient methods, but can yield bad results when applied to actor-critic methods due to a poorly conditioned critic (Zhang & Ma, 2018; Nair et al., 2020).

Offline RL, in a similar grain to imitation learning, also involves learning from a fixed dataset of pre-collected experience generated by an oracle policy Fu et al. (2020); Fujimoto et al. (2019). However, offline RL assumes that each transition in the dataset is labelled with a reward, and aims to learn a policy which always improves upon the oracle policy in state-distributions well covered by the dataset. In a similar manner to initializing a policy with imitation learning, policies can also be initialized using offline RL (Nair et al., 2020; 2018; Hester et al., 2018; Kumar et al., 2020; Sonabend et al., 2020).

A key aspect of these techniques is that the distribution of experience that comes from the oracle policy is fixed beforehand. This can be detrimental when the learning policy requires more information from the oracle policy in regions that are ill represented in the dataset. In contrast, CCGE allows the learning policy to query the oracle policy whenever it is uncertain, allowing for information from the oracle policy to be gathered dynamically.

### 2.2 Rolling In Oracle Policies

Given an oracle policy, it is possible to involve it at every stage when attempting to learn an optimal policy, possibly resulting in better critics for downstream finetuning. One method is to carry out environment rollouts by taking composite actions that are a weighted sum of the oracle's actions and the learning policy's actions (Rosenstein et al., 2004). As training progresses, the weighting is annealed to favour the learning policy's actions over the oracle's. This is vaguely similar to Kullback-Leibler (KL) regularized RL , which integrates an oracle policy by incorporating the oracle's actions into the actor's action distribution using an additional KL loss between the two distributions. This has been proven to work on actor critic methods on a range of RL benchmarks (Nair et al., 2020; Peng et al., 2019; Siegel et al., 2019; Wu et al., 2019). However, this can cause the actor to perform poorly when trying to assign probability mass to deterministic actions (Rudner et al., 2021). Other methods include Jump Start Reinforcement Learning (JSRL), which

uses an oracle policy to step through the environment for a fixed number of steps before the learning policy interacts with the environment (Uchendu et al., 2022). The number of steps is gradually reduced as training progresses, forming a curriculum that can be gradually learned.

These ideas have an explicitly defined *switching point* - a point where the learning policy changes from learning from the oracle policy to learning on its own, or vice versa. Such switching can be counterproductive when the oracle policy is more experienced in regions of the environment it cannot reach on its own — regions where switching never happens. In contrast, CCGE allows the switching to happen whenever the learning policy is uncertain, which can happen at any point during training.

### 2.3  Other Methods for Improving Sample Efficiency

Curriculum learning aims to form a curriculum of different tasks that progressively become more difficult (Bengio et al., 2009) whilst model based RL aims to gain more detailed world knowledge of the environment more efficiently to reduce the number of steps that need to be taken in it (Hafner et al., 2019; Chen et al., 2022; Hafner et al., 2020; Schrittwieser et al., 2020). Adjacent methods include curiosity-driven RL or intrinsic motivation, whereby the agent is driven by an additional intrinsic reward signal Schmidhuber (1991); Pathak et al. (2017). This intrinsic signal can be based on learning progress, exploration, or novel experiences, leading to the motivated exploration of new states.

### 2.4  Epistemic Uncertainty

When optimizing a model, knowing how much further performance could be improved given more training would be obviously helpful. In statistical learning theory, this general quantity is referred to as *epistemic uncertainty* (Matthies, 2007) or model uncertainty, of which there are multiple formal quantifications. This quantity is different from *aleatoric uncertainty* or uncertainty in the data, which generally refers to the variability in the desired target prediction conditioned on a data point (Hüllermeier & Waegeman, 2021).

A naive approach to estimating epistemic uncertainty in machine learning models is to use the variance of model predictions as a proxy metric (Lakshminarayanan et al., 2017; Gal & Ghahramani, 2016). Epistemic uncertainty estimation is widely known in Bayesian RL, which use Monte Carlo Dropout or model ensembles (Ghosh et al., 2021; Osband et al., 2018; Osband, 2016; Lütjens et al., 2019). Other examples use Gaussian processes or deep kernel learning methods to explicitly store aleatoric uncertainty estimates within the replay buffer (Kuss & Rasmussen, 2003; Engel et al., 2005; Xuan et al., 2018) or distributional models aiming to learn a distribution of returns (Bellemare et al., 2017; Dabney et al., 2018). Such methods allow a direct capture of aleatoric uncertainty, allowing epistemic uncertainty to be estimated as a function of model variance.

Evidential regression (Amini et al., 2020) aims to estimate epistemic uncertainty by proposing evidential priors over the data likelihood function. This works very well for data with stationary distributions, notably in supervised learning (Charpentier et al., 2020; 2021; Malinin & Gales, 2018). However, this technique does not extend to non-stationary distributions, a staple in RL.

More recently, Jain et al. (2021) modelled aleatoric and epistemic uncertainties by relating them to the expected prediction errors of the model. This relation is intuitive as it directly predicts how much improvement can be gained given more data and learning capacity. They further demonstrated how to use this prediction in a limited case of curiousity-driven RL, in a similar grain to Moerland et al. (2017); Nikolov et al. (2018).

## 3  Preliminaries

### 3.1  Notation

This work uses the standard MDP definition (Sutton & Barto, 2018) defined by the tuple $\{\mathbf{S}, \mathbf{A}, \rho, r, \gamma\}$ where $\mathbf{S}$ and $\mathbf{A}$ represent the state and action spaces, $\rho(\mathbf{s}_{t+1}|\mathbf{s}_t, \mathbf{a}_t)$ represent the state transition dynamics, $r_t = r(\mathbf{s}_t, \mathbf{a}_t, \mathbf{s}_{t+1})$ represents the reward function and $\gamma \in (0, 1)$ represents the discount factor. An agent interacts with the MDP according to the policy $\pi(\mathbf{a}_t|\mathbf{s}_t)$, and during training, transition tuples of $\{\mathbf{s}_t, \mathbf{a}_t, r_t, \mathbf{s}_{t+1}\}$ are

stored in a replay buffer $\mathcal{D}$. The goal of RL is to find the optimal policy $\pi^*$ that maximizes the cumulative discounted rewards $\mathbb{E}[\sum_{t=0}^{\infty} \gamma^t r_t]$ for any state $s_0 \in \mathbf{S}$, where $\mathbf{a}_t \sim \pi(\mathbf{s}_t)$ and $\mathbf{s}_{t+1} \sim \rho(\mathbf{s}_t, \mathbf{a}_t)$.

## 3.2 Deep Q Network

One of the major approaches to finding an optimal policy is via Q-learning, which aims to learn a state-action value function $Q^\pi : \mathbf{S} \times \mathbf{A} \to \mathbb{R}$, defined as the expected cumulative discounted rewards from taking action $\mathbf{a}_t$ in state $\mathbf{s}_t$. The Deep Q Network (DQN) (Mnih et al., 2015) was one of such approaches, in which transition tuples are sampled from $\mathcal{D}$ to learn the function $Q_\phi^\pi$ parameterized by $\phi$,. Here, the Bellman error is minimized via the loss function:

$$\mathcal{L}_Q(\mathbf{s}_t, \mathbf{a}_t) = \mathbb{E}[l(Q_\phi^\pi(\mathbf{s}_t, \mathbf{a}_t) - r_t - \gamma Q_\phi^\pi(\mathbf{s}_{t+1}, \mathbf{a}_{t+1}))] \tag{1}$$

where $l(\cdot)$ is usually the squared error loss and the expectation is taken over $\{\mathbf{s}_t, \mathbf{a}_t, \mathbf{s}_{t+1}, r_t\} \sim \mathcal{D}, \mathbf{a}_{t+1} \sim \pi(\cdot|\mathbf{s}_{t+1})$. In DQN, the optimal policy is defined as $\pi(\mathbf{a}_t|\mathbf{s}_t) = \delta(\mathbf{a} - \arg\max_{\mathbf{a}_t} Q_\phi^\pi(\mathbf{s}_t, \mathbf{a}_t))$, where $\delta(\cdot)$ is the Dirac function. This loss function in equation 1 will be used in derivations later in this work.

## 3.3 Soft Actor Critic (SAC)

The Soft Actor Critic (SAC) algorithm is an entropy regularized actor critic algorithm introduced by Haarnoja et al. (2018a). This work specifically refers to SAC-v2 (Haarnoja et al., 2018b), a slightly improved variant with twin delayed Q networks and automatic entropy tuning. SAC has a pair of models parameterized by $\theta$ and $\phi$ that represent the actor $\pi_\theta$ and the critic $Q_\phi^\pi$. In a similar grain to DQN, the critic aims to learn the state-action value function via Q-learning. However, here, the critic is modified to include entropy regularization to promote exploration and stability for continuous action spaces. The actor is then optimized by minimizing the following loss function via the critic:

$$\mathcal{L}_\pi(\mathbf{s}_t, \mathbf{a}_t) = -\mathbb{E}_{\substack{\mathbf{s}_t \sim \mathcal{D} \\ \mathbf{a}_t \sim \pi_\theta}}[Q_\phi^\pi(\mathbf{s}_t, \mathbf{a}_t)] \tag{2}$$

Note that we have dropped the entropy term for brevity as it is not relevant to our derivations, but it can be viewed in the formulation by Haarnoja et al. (2018b).

## 3.4 Formalism of Epistemic Uncertainty

We present the formalism for epistemic uncertainty presented by Jain et al. (2021) and the related definitions in the notation that will be used in this paper. Consider a learned function $f$ that tries to minimize the expected value of the loss $l(f(x) - y)$ where $y \sim P(\mathcal{Y}|x)$.

**Definition 1** The *total uncertainty* $\mathcal{U}_f(x)$ of a function $f$ at an input $x$, is defined as the expected loss $l(f(x) - y)$ under the conditional distribution $y \sim P(\mathcal{Y}|x)$.

$$\mathcal{U}_f(x) = \mathbb{E}_{y \sim P(\mathcal{Y}|x)}[l(f(x) - y)] \tag{3}$$

This expected loss stems from the random nature of the data $P(\mathcal{Y}|x)$ (aleatoric uncertainty) as well as prediction errors by the function due to insufficient knowledge (epistemic uncertainty).

**Definition 2** A *Bayes optimal predictor* $f^*$ is defined as the predictor $f$ of sufficient capacity that minimizes $\mathcal{U}_f$ at every point $x$.

$$f^* = \arg\min_f \mathcal{U}_f(x) \tag{4}$$

**Definition 3** The *aleatoric uncertainty* $\mathcal{A}(\mathcal{Y}|x)$ of some data $y \sim P(\mathcal{Y}|x)$ is defined as the irreducible uncertainty of a predictor. It can also be viewed as the total uncertainty of a Bayes optimal predictor.

$$\mathcal{A}(\mathcal{Y}|x) = \mathcal{U}_{f^*}(x) \tag{5}$$

Note that the aleatoric uncertainty is defined over the conditional data distribution, and is not conditioned on the estimator. By definition, $\mathcal{A}(\mathcal{Y}|x) \le \mathcal{U}_f(x), \forall f \forall x$.

When $P(\mathcal{Y}|x)$ is Gaussian and $l(\cdot)$ is defined as the squared error loss, an optimal predictor predicts the mean of the conditional distribution, $f^*(x) = \mathbb{E}_{y \sim P(\mathcal{Y}|x)}[y]$. As a result, the total uncertainty of an optimal predictor is equivalent to the variance of the conditional distribution. Thus, by extension, the aleatoric uncertainty of any predictor on Gaussian data trained using the squared error loss is equal to the variance of the data itself.

$$
\begin{aligned}
\mathcal{A}(\mathcal{Y}|x) &= \mathcal{U}_{f^*}(x) \\
&= \mathbb{E}_{y \sim P(\mathcal{Y}|x)}[l(y - \mathbb{E}_{y \sim P(\mathcal{Y}|x)}[y])] \\
&= \mathbb{E}_{y \sim P(\mathcal{Y}|x)}[l(y)] - l(\mathbb{E}_{y \sim P(\mathcal{Y}|x)}[y]) \\
&= \sigma_{|x}^2(\mathcal{Y})
\end{aligned}
\tag{6}
$$

Note that we have chosen to write $\sigma^2(\mathcal{Y}|x)$ as $\sigma_{|x}^2(\mathcal{Y})$ for brevity.

**Definition 5** The *epistemic uncertainty* $\mathcal{E}_f(x)$ is defined as the difference between total uncertainty and aleatoric uncertainty. This quantity will asymptotically approach zero as the amount of data goes to infinity for a predictor $f$ with sufficient capacity.

$$
\mathcal{E}_f(x) = \mathcal{U}_f(x) - \mathcal{A}(\mathcal{Y}|x) = \mathcal{U}_f(x) - \mathcal{U}_{f^*}(x)
\tag{7}
$$

The right hand side of equation 7 provides an intuitive view of epistemic uncertainty — it is the difference in performance between the current model and an optimal one.

## 4    Critic Confidence Guided Exploration

In this section, we propose a technique to improve the sample efficiency of RL agents by choosing to mimic an oracle policy when uncertain, and performing self-exploration when the oracle policy's actions have been explored. In contrast to current techniques where the agent has no control over when to mimic the oracle policy (e.g. annealed weighting in Rosenstein et al. (2004), random oracle policy walks in Uchendu et al. (2022)), our technique leverages the uncertainty of the agent to control when to follow the oracle policy. We term this algorithm CCGE, in reference to the learning policy mimicking the oracle policy for exploration when the confidence in the $Q$-value estimate is low.

### 4.1    Incorporating an Oracle Policy

Our method of improving sample efficiency with an oracle policy is loosely inspired by the Upper Confidence Bound (UCB) Bandit algorithm. Assume that we have a critic $Q_\phi^\pi$ and means of estimating its epistemic uncertainty $\mathcal{E}_\phi$. For any state and action $\{\mathbf{s}_t, \mathbf{a}_t\}$, we can assign an upper-bound to the true $Q^\pi$ value:

$$
Q_{\mathrm{UB}}^\pi(\mathbf{s}_t, \mathbf{a}_t) = Q_\phi^\pi(\mathbf{s}_t, \mathbf{a}_t) + \mathcal{E}_\phi(\mathbf{s}_t, \mathbf{a}_t)
\tag{8}
$$

In a state $\mathbf{s}_t$, the potential improvement of following the oracle policy's suggested action $\bar{\mathbf{a}}_t$ can be then canonically written as:

$$
\Delta(\mathbf{s}_t, \mathbf{a}_t, \bar{\mathbf{a}}_t) = Q_{\mathrm{UB}}^\pi(\mathbf{s}_t, \bar{\mathbf{a}}_t) - Q_{\mathrm{UB}}^\pi(\mathbf{s}_t, \mathbf{a}_t)
\tag{9}
$$

The term $Q_{\mathrm{UB}}^\pi(\mathbf{s}_t, \bar{\mathbf{a}}_t)$ refers to the upper-bound $Q$-value when taking the action $\bar{\mathbf{a}}_t$ and then acting according to the learning policy $\pi_\theta$. Now, the decision to learn from the oracle policy versus the critic can be made by defining the actor's loss function as either a supervisory signal $\mathcal{L}_{\mathrm{sup}}$ or a reinforcement signal $\mathcal{L}_\pi$ (from equation 2):

$$
\mathcal{L}_{\mathrm{CCGE}}(\mathbf{s}_t, \mathbf{a}_t, \bar{\mathbf{a}}_t) = \begin{cases} \mathcal{L}_{\mathrm{sup}}(\mathbf{s}_t, \mathbf{a}_t, \bar{\mathbf{a}}_t), & \text{if } k \geq \lambda \\ \mathcal{L}_\pi(\mathbf{s}_t, \mathbf{a}_t), & \text{otherwise} \end{cases}
\tag{10}
$$

where $k$ is computed with:

$$
k = \frac{\Delta(\mathbf{s}_t, \mathbf{a}_t, \bar{\mathbf{a}}_t)}{Q_\phi^\pi(\mathbf{s}_t, \mathbf{a}_t)}
\tag{11}
$$

and $\lambda$ is a constant which we term *confidence scale*. To put simply, the choice between imitating the oracle policy versus performing reinforcement learning is chosen according to the normalized potential improvement

of doing the former versus the latter. In general, CCGE is flexible to choices of $\lambda$, and we show preliminary results of different values of $\lambda$ in Appendix B.

During training policy rollout, the learning policy may also choose to take the action from the oracle policy:

$$\mathbf{a}_t \leftarrow \begin{cases} \bar{\mathbf{a}}_t, \text{if } k \geq \lambda \\ \mathbf{a}_t, \text{otherwise} \end{cases} \tag{12}$$

This allows the learning policy to quickly see the effect of the oracle policy's suggestions, helpful when the learning policy is completely uncertain about its environment at the beginning of training. We do not do this step at inference or policy evaluation.

## 4.2 Supervision Signal Definition

There are various ways to define the supervision loss $\mathcal{L}_{\mathrm{sup}}$. For continuous action spaces, we can simply use the squared error loss:

$$\mathcal{L}_{\mathrm{sup}}(\mathbf{s}_t, \mathbf{a}_t, \bar{\mathbf{a}}_t) = \mathbb{E}_{\substack{\mathbf{a}_t \sim \pi_\theta(\cdot|\mathbf{s}_t) \\ \bar{\mathbf{a}}_t \sim \bar{\pi}(\cdot|\mathbf{s}_t)}}[||\mathbf{a}_t - \bar{\mathbf{a}}_t||_2^2] \tag{13}$$

That said, any other loss function in similar contexts can be used, such as minimizing the negative log likelihood $-\log \pi_\theta(\bar{\mathbf{a}}_t)$, $L1$ loss function $|\mathbf{a}_t - \bar{\mathbf{a}}_t|$, or similar. Discrete action space models can instead utilize the cross entropy loss $-\bar{\mathbf{a}}_t \log(\pi_\theta(\mathbf{a}_t))$.

## 4.3 Epistemic Uncertainty Metrics for a Critic

We present two metrics for quantifying the epistemic uncertainty of a critic — one based on $Q$-network ensembles in a similar grain to past work (Osband, 2016; Clements et al., 2019; Festor et al., 2021) which we call Implicit Epistemic Uncertainty, and one based on the DEUP technique (Jain et al., 2021) which we call Explicit Epistemic Uncertainty. We do not argue which metric provides a more holistic estimate of epistemic uncertainty, interested readers are instead directed to work by Charpentier et al. (2022) for a more detailed study. Instead, CCGE simply assumes a means of evaluating the epistemic uncertainty of the $Q$-value estimate given a state action pair, and these are two such examples.

### 4.3.1 Implicit Epistemic Uncertainty

We adopt the simplest form of estimating epistemic uncertainty in this regime using an ensemble of $Q$-networks. In SAC, an ensemble of $n(=2)$ $Q$-networks are used to tame overestimation bias (Hasselt, 2010; Haarnoja et al., 2018b). For every state action pair, a set of $Q$-value estimates, $\{Q_{\phi_1}^\pi, Q_{\phi_2}^\pi, ...Q_{\phi_n}^\pi\}$ are obtained. We utilize the variance of these $Q$-values as a proxy metric for epistemic uncertainty $\mathcal{E}_\phi = \sigma^2(\{Q_{\phi_1}^\pi, Q_{\phi_2}^\pi, ...Q_{\phi_n}^\pi\})$ and set $Q_\phi^\pi = \min(Q_\phi^1, Q_\phi^2, ...Q_\phi^n)$ as is done in the original SAC implementation to obtain $k$.

### 4.3.2 Explicit Epistemic Uncertainty

Adopting the framework for epistemic uncertainty from Jain et al. (2021), we derive an estimate for epistemic uncertainty based on the Bellman residual error. More formally, we first define single step epistemic uncertainty for a $Q$-value estimate as:

$$\begin{aligned} \delta_t(\mathbf{s}_t, \mathbf{a}_t) &= \mathcal{U}_\phi(\mathbf{s}_t, \mathbf{a}_t) - \mathcal{A}(Q_\phi^\pi|\{\mathbf{s}_t, \mathbf{a}_t\}) \\ &= l\big(\mathbb{E}[Q_\phi^\pi(\mathbf{s}_t, \mathbf{a}_t) - r_t - \gamma Q_\phi^\pi(\mathbf{s}_{t+1}, \mathbf{a}_{t+1})]\big) \end{aligned} \tag{14}$$

This is simply the expected Bellman residual error projected through the squared error loss. The exact derivation is available in Appendix A.1.

Then, we propose using the root of the discounted sum of single step epistemic uncertainties as a measure for total epistemic uncertainty given a state and action pair:

$$\mathcal{E}_\phi(\mathbf{s}_t, \mathbf{a}_t) = \left[\mathbb{E}_{\pi, \mathcal{D}}\left[\sum_{i=t}^T \gamma^{i-t}|\delta_i(\mathbf{s}_i, \mathbf{a}_i)|\right]\right]^{\frac{1}{2}} \tag{15}$$

The intuition here is that $\delta_t$ is a biased estimate of epistemic uncertainty. To obtain a more holistic view of epistemic uncertainty, we instead take the discounted sum of all single step epistemic uncertainty estimates as the true measure. In our experiments, we found that taking the root of this value results in more stable training dynamics.

The value of $\mathcal{E}_\phi$ can be estimated on an auxiliary output of the $Q$-network itself, and learned by minimizing the residual loss in equation 16:

$$\mathcal{L}_\mathcal{E}(\mathbf{s}_t, \mathbf{a}_t) = l(\mathcal{E}_\phi(\mathbf{s}_t, \mathbf{a}_t) - \left(\delta_t(\mathbf{s}_t, \mathbf{a}_t) + \gamma\mathcal{E}_\phi(\mathbf{s}_{t+1}, \mathbf{a}_{t+1})^2\right)^{\frac{1}{2}}) \tag{16}$$

The resulting loss function for the $Q$-value network for a single state action pair can then be derived from equation equation 1 and equation 16:

$$\mathcal{L}_{\mathcal{E},Q}(\mathbf{s}_t, \mathbf{a}_t) = \mathcal{L}_\mathcal{E}(\mathbf{s}_t, \mathbf{a}_t) + \mathcal{L}_Q(\mathbf{s}_t, \mathbf{a}_t) \tag{17}$$

### 4.4 Algorithm

We present the full pseudocode for CCGE in Algorithm 1. The standard parameters for an actor-critic reinforcement learning algorithm are first initialized. Then, we select a confidence scale that controls how confident the algorithm should be about the learning policy's actions. For most tasks, a value of 1 has been found to work well, while smaller values near 0.1 may result in better performance for harder exploration tasks. During environment rollout, actions are taken in the environment according to the oracle policy or learning policy depending on $k$. Similarly, during policy optimization, the learning policy either performs supervised learning against the oracle policy's suggested actions or standard reinforcement learning depending on $k$. In Algorithm 1, `UpdateCritic` is the standard reinforcement learning value function update, usually done by minimizing equation 1 when using Implicit Epistemic Uncertainty, or equation 17 for Explicit Epistemic Uncertainty.

---

**Algorithm 1** Critic Confidence Guided Exploration (CCGE)

    Select discount factor $\gamma$, learning rates $\eta_\phi$, $\eta_\pi$
    Select size of $Q$-value network ensemble $n$
    Select confidence scale $\lambda \geq 0$

    Initialize parameter vectors $\theta$, $\phi$
    Initialize actor and critic networks $\pi_\theta$, $Q_\phi$
    Initialize or hardcode oracle $\bar{\pi}$

    **for** number of episodes **do**
        Initialize $\mathbf{s}_t = \mathbf{s}_{t=0} \sim P(\cdot)$
        **while** env not done **do**
            Sample $\mathbf{a}_t$ from $\pi_\theta(\cdot|\mathbf{s}_t)$
            Sample $\bar{\mathbf{a}}_t$ from $\bar{\pi}(\cdot|\mathbf{s}_t)$
            Compute $k$ using $Q_\phi(\mathbf{s}_t, \mathbf{a}_t), Q_\phi(\mathbf{s}_t, \bar{\mathbf{a}}_t), \mathcal{E}(\mathbf{s}_t, \mathbf{a}_t), \mathcal{E}_\phi(\mathbf{s}_t, \bar{\mathbf{a}}_t)$
            Override $\mathbf{a}_t \leftarrow \bar{\mathbf{a}}_t$ if $k \geq \lambda$
            Sample $r_t, \mathbf{s}_{t+1}$ from $\rho(\cdot|\mathbf{s}_t, \mathbf{a}_t)$
            Store transition tuples $\{\mathbf{s}_t, \mathbf{a}_t, r_t, \mathbf{s}_{t+1}, \bar{\mathbf{a}}_t\}$ in $\mathcal{D}$
        **end while**

        **for** $\{\mathbf{s}_t, \mathbf{a}_t, r_t, \mathbf{s}_{t+1}, \bar{\mathbf{a}}_t\}$ **in** $\mathcal{D}$ **do**
            Update critic $Q_\phi \leftarrow$ `UpdateCritic`$(\mathbf{s}_t, \mathbf{a}_t, r_t, Q_\phi)$
            Update actor parameters $\theta \leftarrow \theta - \eta_\phi\nabla_\phi\mathcal{L}_{\text{CCGE}}$
        **end for**
    **end for**

---

# 5 Experiments

We implement CCGE on SAC, specifically SACv2 (Haarnoja et al., 2018b), and then evaluate its behaviour and performance on a variety of environments against a range of other algorithms. We use environments from Gymnasium (formerly Gym) (Brockman et al., 2016), D4RL derived environments from Gymnasium Robotics (Fu et al., 2020), and waypoint environments from PyFlyt (Tai & Wong, 2023). Following guidelines from Agarwal et al. (2021), we report 50% Interquartile Means (IQMs) with bootstrapped confidence intervals. For each configuration, we train for 1 million environment timesteps and aggregate results based on 50 seeds per configuration and calculate evaluation scores based on 100 rollouts every 10,000 timesteps. The results of our experiments are shown in Figure 1 and Figure 2 with all relevant hyperparameters and network setups recorded in Appendix D, E, and F. The experiments are aimed at answering the following questions:

1. Does CCGE benefit most from the supervision signal or the guidance from actions taken by the oracle?

2. How does CCGE compare over the baseline algorithm with no supervision?

3. How important is the performance of the oracle policy towards the performance of CCGE?

4. Is CCGE sensitive to different methods of epistemic uncertainty estimation?

5. Can we bootstrap the oracle using the learning policy continuously?

6. How does CCGE perform on hard exploration tasks in comparison to other state of the art algorithms?

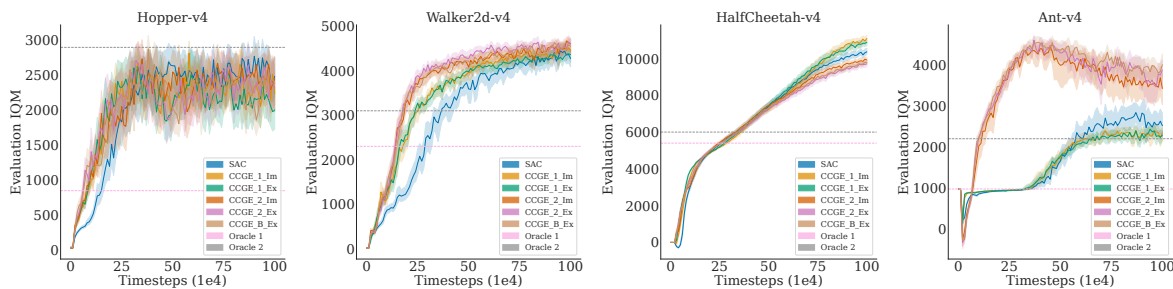

Figure 1: Learning curves of CCGE and SAC on Gym Mujoco environments. For the CCGE runs, the run names are written as `CCGE_{Oracle Number}_{Epistemic Uncertainty Estimation Type}`. The oracle policy labelled `B` denotes the bootstrapped oracle policy.

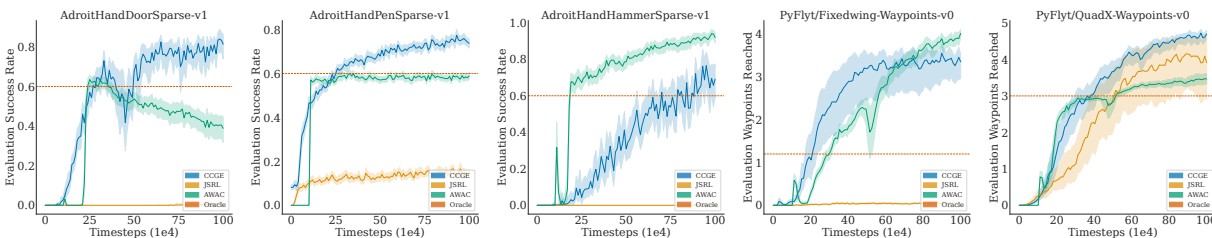

Figure 2: Learning curves of CCGE, AWAC and JSRL on three Gymnasium Robotics and two PyFlyt environments.

## 5.1 Supervision or Guidance

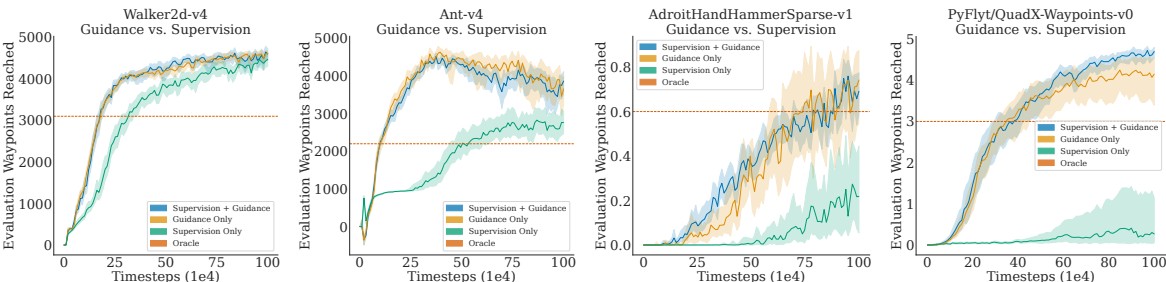

Figure 3: Learning curves of CCGE, AWAC and JSRL on three Gymnasium Robotics and two PyFlyt environments.

Algorithmically, CCGE uses two techniques to incorporate the oracle policy — a supervision signal induced during the policy improvement phase, and guidance via taking the oracle's actions directly during policy rollout. We conduct a series of experiments using Walker-v4, Ant-v4, PyFlyt/QuadX-Waypoints-v0 and AdroitHandHammerSparse-v1 with either only guidance, only supervision, or both enabled to determine the effect that either component has on CCGE's performance. The results are shown in Figure 3. In all cases, using the supervision only variant of CCGE produces the worst performance, with performance similar to the base SAC algorithm in the case of Walker2d-v4 and Ant-v4. CCGE performance benefits most from oracle guidance, with the guidance only algorithm performing similarly to the full algorithm in all dense reward environments. For sparse reward environments, the supervision component in the full algorithm produces a minimal but non-negligible learning improvement in terms of final performance in PyFlyt/QuadX-Waypoints-v0 and initial rate of improvement in AdroitHandHammerSparse-v1.

## 5.2 CCGE Ablations

To answer questions 2, 3, and 4, we use the continuous control, dense reward, robotics tasks from the MuJoCo suite of Gymnasium environments (Brockman et al., 2016; Todorov et al., 2012). More concisely, we use Hopper-v4, Walker2d-v4, HalfCheetah-v4 and Ant-v4. These environments are canonically similar to those used by Haarnoja et al. (2018b) and various other works. For each environment, we obtain two oracles — Oracle 1 and Oracle 2 — using SAC trained for $250 \times 10^3$ and $500 \times 10^3$ respectively. Both oracles have different final performances that are generally lower than what is obtainable using SAC trained to convergence. For each oracle, we evaluate the performance of CCGE using two separate methods of estimating epistemic uncertainty — the implicit and explicit methods detailed in Sections 4.3.1 and 4.3.2. This results in a total of four CCGE runs per environment, which we use to compare results against SAC. We label CCGE with the first oracle as `CCGE_1` and with the second, better performing oracle as `CCGE_2`. To denote between different forms of epistemic uncertainty estimation, we further append either `_Im` or `_Ex` to the algorithm names to denote usage of either the implicit or explicit forms of epistemic uncertainty estimation.

### 5.2.1 CCGE vs. no CCGE

Compared against the baseline SAC algorithm, CCGE outperforms SAC in all cases using the right oracle policy. In some cases, CCGE significantly outperforms SAC by leveraging the oracle policy to escape local minima early on. The obvious example here is in the Ant-v4 environment. Here, SAC tends to learn a standing configuration very early on, achieving an evaluation score of about 1000. It takes approximately $300 \times 10^3$ more transitions before the algorithm achieves a stable walking policy that does not fall. Using the right oracle policy — in this case a mostly walking oracle with an evaluation score of 2100, CCGE learns a successful walking configuration in approximately 15% the number of transitions that it takes for SAC. The performance of CCGE in the Ant-v4 environment seems to degrade over time. One possible reason for this

could be catastrophic forgetting due to the limited capacity replay buffer (Isele & Cosgun, 2018), and this is explored more in Appendix C.

### 5.2.2 Oracle Performance Sensitivity

Most reinforcement learning algorithms that bootstrap off imitation learning are improved by having higher quality data. CCGE is no exception to this dependency, as the change in performance from having different oracle policies directly dictates the learning performance of the algorithm. In particular, a different oracle policy for HalfCheetah-v4 results in CCGE either performing better or worse than SAC. Similarly, when initialized with an oracle policy stuck in a local minima in Ant-v4, CCGE causes the learning policy to learn slightly slower than SAC due to it repeatedly reverting to the suboptimal oracle policy when uncertain. Despite this, the learning policy escapes the local minima in about the same time as SAC. This suggests that while CCGE with a bad oracle policy can hamper learning performance, it does not limit the exploration rate of the learning policy in general.

### 5.2.3 Different Epistemic Uncertainty Estimation Techniques

We use the same confidence scale $\lambda$ (introduced in Section 4.1) for both implementations. The results from Figure 1 show that while choice of epistemic uncertainty estimation technique does impact learning performance, the impact is far smaller than a change of oracle policy. More concretely, implicit epistemic uncertainty estimation works slightly better in Walker2d-v4 and HalfCheetah-v4, but performs worse in Ant-v4. As long as CCGE has access to a reasonable measure of epistemic uncertainty, CCGE would benefit much more from better performing oracle policies than better epistemic uncertainty estimation techniques.

### 5.3 Bootstrapped Oracle

One advantage of CCGE's formulation is that it does not make any assumptions about the oracle policy. In fact, it does not even require the state-conditioned distribution of actions from the oracle policy to be stationary as required by algorithms such as Implicit Q Learning (IQL) (Kostrikov et al., 2023). Therefore, CCGE can function even when the oracle policy is continuously improving. To test this theory, we evaluated a variant of CCGE where the oracle policy's weights are updated to match the learning policy's weights whenever the learning policy's evaluation performance surpassed that of the oracle policy. We use CCGE with explicit epistemic uncertainty estimation, with the same MuJoCo suite setup as done previously and the oracle policy initialized using Oracle 2. The resulting runs are labelled as `CCGE_B_Ex`. Surprisingly, we found that using a bootstrapped oracle in this scenario did not provide any meaningful improvement in performance.

### 5.4 Hard Exploration Tasks

We evaluate the performance of CCGE on hard exploration tasks against Advantage Weighted Actor Critic (AWAC) and JSRL using an SAC backbone — two algorithms suitable for this setting of learning from prior data in conjunction with environment interaction. Their performances are evaluated on the AdroitHandDoorSparse-v1, AdroitHandPenSparse-v1 and AdroitHandHammerSparse-v1 tasks from Gymnasium Robotics, as well as the Fixedwing-Waypoints-v0 and QuadX-Waypoints-v0 tasks from PyFlyt. Hard exploration tasks describe environments where rewards are flat everywhere except when a canonical goal has been achieved. For example, in AdroitHandDoorSparse-v1 from Gymnasium Robotics, the reward is -0.1 at every timestep and 10 whenever the goal of opening the door completely has been achieved. Similarly, in PyFlyt the reward is -0.1 at every timestep and 100 when the agent has reached a waypoint.

The oracles for the AdroidHand tasks were obtained using SAC trained in the dense reward setup until the evaluation performance reached 0.6, requiring about $200 \times 10^3$ to $400 \times 10^3$ environment steps in the dense reward setup. The oracle for Fixedwing-Waypoints-v0 was obtained similarly — using SAC until the agent reaches 3 waypoints during evaluation, which happened after approximately $300 \times 10^3$ environment steps. For QuadX-Waypoints-v0, a cascaded Proportional Integral Derivative (PID) controller (Kada & Ghazzawi,

2011) which partially solves this environment is deployed as the oracle policy. The variant of CCGE here uses explicit epistemic uncertainty estimation.

When compared with other competitive methods on the Gymnasium Robotics and PyFlyt tasks, CCGE demonstrates competitive sample efficiency and final performance on most tasks. This implies that CCGE can effectively explore and learn from environments with sparse rewards, which can be challenging for traditional RL algorithms. Notably, CCGE excels on environments that require more exploration than incremental improvement, such as AdroitHandPenSparse-v1, AdroitHandDoorSparse-v1, and PyFlyt/QuadX-Waypoints-v0. Its underlying SAC's maximum entropy paradigm allows for aggressive exploration, which enables CCGE to quickly surpass the performance of the oracle policy. This is in contrast to AWAC which struggles to perform much better than the oracle policy due to more conservative exploration. However, on environments with a more narrow range of optimal action sequences, such as AdroitHandHammerSparse-v1 and Fixedwing-Waypoints-v0, AWAC's more conservative policy updates result in better overall learning performance and final performance. Interestingly, JSRL fails to perform well in most tasks except one. This could be due to a few factors, such as its reliance on the underlying IQL backbone or the potential idea that simply starting from good starting states is insufficient in guaranteeing good performance.

## 6 Conclusion

This paper introduces a new approach for RL called Critic Confidence Guided Exploration (CCGE), which seeks to address the challenges of exploration during optimization. The approach bootstraps from an oracle policy and uses the critic's prediction error or variance as a proxy for uncertainty to determine when to learn from the oracle and when to optimize the learned value function. We empirically evaluate CCGE on several benchmark robotics tasks from Gymnasium, evaluating its learning behaviour under different oracle policies and different methods of measuring uncertainty. We further test CCGE's performance on various sparse reward environments from Gymnasium Robotics and PyFlyt and find that CCGE offers competitive sample efficiency and final performance on most tasks against other, well performing algorithms, such as Advantage Weighted Actor Critic (AWAC) and Jump Start Reinforcement Learning (JSRL). Notably, CCGE excels on environments that require more exploration than incremental improvement, while other methods perform better on tasks with a more narrow range of optimal action sequences. We expect that CCGE will motivate more research in this direction, where leveraging an oracle in reinforcement learning is guided by an intrinsic heuristic. In the future, we would like to explore scenarios with multiple oracle policies, as well as the potential of utilizing CCGE in a multi-agent setting, where agents utilize other agents in the environment as oracles.

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

## A    Explicit Epistemic Uncertainty

In this section, we derive epistemic uncertainty in terms of $Q$-value networks for any $\{\mathbf{s}_t, \mathbf{a}_t\}$ pair. This derivation follows the formalism taken from Jain et al. (2021) briefly covered in Section 3.4. Unless otherwise specified, all expectations in this section are taken over $\mathbf{s}_{t+1}, r_t \sim \rho_{\text{env}}(\cdot|\mathbf{s}_t, \mathbf{a}_t), \mathbf{a}_{t+1} \sim \pi_\theta(\cdot|\mathbf{s}_{t+1})$.

### A.1    Single Step Epistemic Uncertainty

For a $Q$-value estimator, following the definition in equation 3, the *single step total uncertainty* can be written as the expected total loss for $Q_\phi^\pi$:

$$\mathcal{U}_\phi(\mathbf{s}_t, \mathbf{a}_t) = \mathbb{E}\big[l(Q_\phi^\pi(\mathbf{s}_t, \mathbf{a}_t) - r_t - \gamma Q_\phi^\pi(\mathbf{s}_{t+1}, \mathbf{a}_{t+1}))\big] \tag{18}$$

The expectation here is taken over $\{\mathbf{s}_{t+1}, r_t\} \sim \mathcal{D}|\{\mathbf{s}_t, \mathbf{a}_t\}, \mathbf{a}_{t+1} \sim \pi(\cdot|\mathbf{s}_{t+1})$. Likewise, we can define a *single step aleatoric uncertainty* for an estimated $Q$-value by extending equation 5 and equation 6. Recall that the aleatoric uncertainty is defined over the target distribution (in this case $\mathbb{E}[r_t + \gamma Q_\phi^\pi(\mathbf{s}_{t+1}, \mathbf{a}_{t+1})]$), and assuming that the target distribution is Gaussian, the aleatoric uncertainty simply becomes the variance in the data. Using this assumption, we can write the aleatoric uncertainty as the variance of the target $Q$-value estimate.

$$\mathcal{A}(Q_\phi^\pi|\{\mathbf{s}_t, \mathbf{a}_t\}) = \sigma_{|\mathbf{s}_t, \mathbf{a}_t}^2 \left(r_t + Q_\phi^\pi(\mathbf{s}_{t+1}, \mathbf{a}_{t+1})\right) \tag{19}$$

Finally, we denote the epistemic uncertainty for $Q$-value estimates across one time step for a given state and action as $\delta_t(\mathbf{s}_t, \mathbf{a}_t)$. Following equation equation 7 and using equation 18, this is defined as:

$$\begin{aligned}
&\delta_t(\mathbf{s}_t, \mathbf{a}_t) \\
&= \mathcal{U}_\phi(\mathbf{s}_t, \mathbf{a}_t) - \mathcal{A}(Q_\phi^\pi|\{\mathbf{s}_t, \mathbf{a}_t\}) \\
&= \mathbb{E}\big[l(Q_\phi^\pi(\mathbf{s}_t, \mathbf{a}_t) - r_t - \gamma Q_\phi^\pi(\mathbf{s}_{t+1}, \mathbf{a}_{t+1}))\big] - \sigma_{|\mathbf{s}_t, \mathbf{a}_t}^2 \left(r_t + Q_\phi^\pi(\mathbf{s}_{t+1}, \mathbf{a}_{t+1})\right)
\end{aligned} \tag{20}$$

Taking the definition of variance as $\sigma^2(x) = \mathbb{E}[l(x)] - l(\mathbb{E}[x])$, the first term in equation 20 can be expanded into:

$$
\begin{aligned}
\mathbb{E}&\big[l(Q_\phi^\pi(\mathbf{s}_t, \mathbf{a}_t) - r_t - \gamma Q_\phi^\pi(\mathbf{s}_{t+1}, \mathbf{a}_{t+1}))\big] \\
&= \sigma^2_{|\mathbf{s}_t, \mathbf{a}_t}(Q_\phi^\pi(\mathbf{s}_t, \mathbf{a}_t) - r_t - Q_\phi^\pi(\mathbf{s}_{t+1}, \mathbf{a}_{t+1})) + l\big(\mathbb{E}[Q_\phi^\pi(\mathbf{s}_t, \mathbf{a}_t) - r_t - \gamma Q_\phi^\pi(\mathbf{s}_{t+1}, \mathbf{a}_{t+1})]\big)
\end{aligned}
\tag{21}
$$

Since $\sigma^2(Q_\phi^\pi(\mathbf{s}_t, \mathbf{a}_t)) = 0$, $\sigma^2(x + C) = \sigma^2(x)$ for constant $C$, and $\sigma^2(-x) = \sigma^2(x)$, equation 21 evaluates to:

$$
\begin{aligned}
\mathbb{E}&\big[l(Q_\phi^\pi(\mathbf{s}_t, \mathbf{a}_t) - r_t - \gamma Q_\phi^\pi(\mathbf{s}_{t+1}, \mathbf{a}_{t+1}))\big] \\
&= \sigma^2_{|\mathbf{s}_t, \mathbf{a}_t}(Q_\phi^\pi(\mathbf{s}_t, \mathbf{a}_t) - r_t - Q_\phi^\pi(\mathbf{s}_{t+1}, \mathbf{a}_{t+1})) + l\big(\mathbb{E}[Q_\phi^\pi(\mathbf{s}_t, \mathbf{a}_t) - r_t - \gamma Q_\phi^\pi(\mathbf{s}_{t+1}, \mathbf{a}_{t+1})]\big) \\
&= \sigma^2_{|\mathbf{s}_t, \mathbf{a}_t}(-r_t - Q_\phi^\pi(\mathbf{s}_{t+1}, \mathbf{a}_{t+1})) + l\big(\mathbb{E}[Q_\phi^\pi(\mathbf{s}_t, \mathbf{a}_t) - r_t - \gamma Q_\phi^\pi(\mathbf{s}_{t+1}, \mathbf{a}_{t+1})]\big) \\
&= \sigma^2_{|\mathbf{s}_t, \mathbf{a}_t}(r_t + Q_\phi^\pi(\mathbf{s}_{t+1}, \mathbf{a}_{t+1})) + l\big(\mathbb{E}[Q_\phi^\pi(\mathbf{s}_t, \mathbf{a}_t) - r_t - \gamma Q_\phi^\pi(\mathbf{s}_{t+1}, \mathbf{a}_{t+1})]\big)
\end{aligned}
\tag{22}
$$

Putting equation 22 into equation 20, we obtain:

$$
\begin{aligned}
\delta_t(\mathbf{s}_t, \mathbf{a}_t) &= \mathcal{U}_\phi(\mathbf{s}_t, \mathbf{a}_t) - \mathcal{A}(Q_\phi^\pi | \{\mathbf{s}_t, \mathbf{a}_t\}) \\
&= l\big(\mathbb{E}[Q_\phi^\pi(\mathbf{s}_t, \mathbf{a}_t) - r_t - \gamma Q_\phi^\pi(\mathbf{s}_{t+1}, \mathbf{a}_{t+1})]\big)
\end{aligned}
\tag{23}
$$

Simply put, the single step epistemic uncertainty for a $Q$ value estimator is simply the expected Bellman residual error projected through the loss function.

## A.2 N-Step Epistemic Uncertainty

The single step epistemic uncertainty is only a measure of epistemic uncertainty of the $Q$-value predicted against its target value. For RL methods which rely on bootstrapping, the target value consists of a sampled reward and an estimated $Q$-value of the next time step. To obtain a more reliable estimate for epistemic uncertainty, it is important to account for the epistemic uncertainty in the target value itself. This train of thought leads very naturally to estimating epistemic uncertainty using the discounted sum of single step epistemic uncertainties, which we refer to as $\mathcal{E}_\phi$.

$$
\mathcal{E}_\phi(\mathbf{s}_t, \mathbf{a}_t) = \mathbb{E}_{\pi, \mathcal{D}}\left[\sum_{i=t}^{T} \gamma^{i-t} |\delta_i(\mathbf{s}_i, \mathbf{a}_i)|\right]
\tag{24}
$$

## A.3 Learning the N-Step Epistemic Uncertainty

In our experiments, we found that having neural networks learn equation 24 leads to very unstable training due to value blowup. Instead, we propose simply learning its root, resulting in much more stable training:

$$
\mathcal{E}_\phi(\mathbf{s}_t, \mathbf{a}_t) = \left[\mathbb{E}_{\pi, \mathcal{D}}\left[\sum_{i=t}^{T} \gamma^{i-t} |\delta_i(\mathbf{s}_i, \mathbf{a}_i)|\right]\right]^{\frac{1}{2}}
\tag{25}
$$

This quantity can be learnt via bootstrapping, in the normal fashion that $Q$-value estimators are learnt using the following recursive sum:

$$
\mathcal{E}_\phi(\mathbf{s}_t, \mathbf{a}_t) = \big(\delta_t(\mathbf{s}_t, \mathbf{a}_t) + \gamma(\mathcal{E}_\phi(\mathbf{s}_{t+1}, \mathbf{a}_{t+1}))^2\big)^{\frac{1}{2}}
\tag{26}
$$

Equation 26 provides a learnable metric for epistemic uncertainty that can be learned via bootstrapping for $Q$-value networks.

One detail is that equation 23 requires taking the expectation of the Bellman residual error projected through the squared error loss. In practice, except for the simplest of environments, this is not entirely possible as it requires access to the reward and next state distribution for given state and action pair. In our experiments, we take the expectation of single state transition samples, and note that this results in inflated estimates of epistemic uncertainty. More concisely, the resulting learnt quantity is much closer to equation 18.

### A.4 Explicit Epistemic Uncertainty Estimates on Gymnasium Environments

We implement DQN with explicit epistemic uncertainty estimation on four Gym environments with increasing difficulty and complexity (Brockman et al., 2016): CartPole, Acrobot, MountainCar, and LunarLander. The goal is to study how this measurement behaves throughout the learning process. Note that **we do not use this measurement of epistemic uncertainty to motivate exploration or mitigate risk as in other works**, the goal is to simply study its behaviour during a standard training run of DQN. We aggregate results over 150 runs using varying hyperparameters shown in Table 1.

On CartPole in Fig. 4, the epistemic uncertainty starts small, increases and then decreases, while evaluation performance exhibits a mostly upward trend. This is perhaps expected, since state diversity –and therefore reward diversity– starts small and increases during training, model uncertainty follows the same initial trend. Eventually, model uncertainty falls as $Q$-value predictions get more accurate through sufficient exploration and $Q$-value network updates, leading to better performance and lower model uncertainties. This trend is similarly observed in Acrobot and MountainCar, albeit to a lesser extent. Conversely, it is not observed for the more challenging LunarLander, where the trend of the $F$-value increases monotonically and plateaus out in aggregate.

Analyzing results from individual runs reveals that in environments with rewards that vary fairly smoothly such as CartPole and LunarLander (Figure 5(a)), the $F$-value can show either performance collapse as in the example shown in CartPole, or indicate state exploration activity as in LunarLander. In environments with sparse rewards such as Acrobot and MountainCar (Figure 5(b)), the $F$-value is indicative of agent learning progress: we observe spikes in uncertainty when the model discovers crucial checkpoints in the environment. For Acrobot, this occurs when the upright position is reached and the reward penalty is stopped. In MountainCar, this occurs when the agent first reaches the top of the mountain, which completes the environment. More examples of similar plots are shown in Appendix A.5.

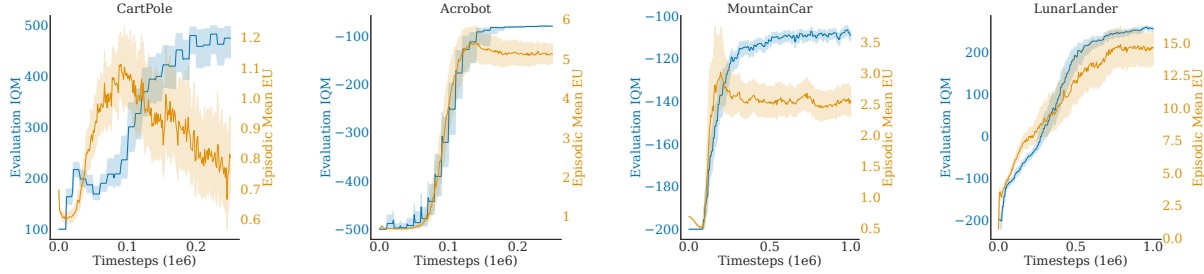

Figure 4: Aggregate episodic mean epistemic uncertainty and evaluation scores across four environments using DQN.

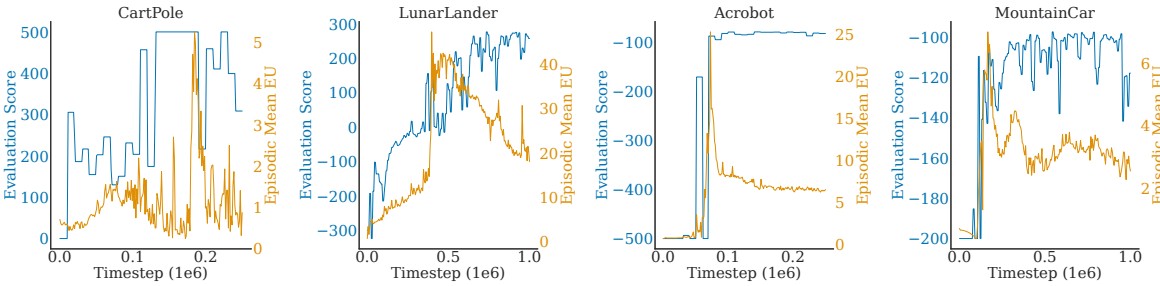

Figure 5: Examples of episodic mean epistemic uncertainty behaviour on non-sparse reward environments using DQN.

Table 1: DQN Hyperparameters for CartPole, Acrobot, MountainCar and Lunarlander

| Parameter | Value |
|---|---|
| *Constants* | |
|   optimizer | AdamW (Loshchilov & Hutter, 2018; Kingma & Ba, 2015) |
|   number of hidden layers (all networks) | 2 |
|   number of neurons per layer (all networks) | 64 |
|   non-linearity | *ReLU* |
|   number of evaluation episodes | 50 |
|   evaluation frequency | every $10 \times 10^3$ steps |
|   *total environment steps* | |
|     CartPole | $250 \times 10^3$ |
|     Acrobot | $250 \times 10^3$ |
|     MountainCar | $1 \times 10^6$ |
|     LunarLander | $1 \times 10^6$ |
|   *replay buffer size* | |
|     CartPole | $100 \times 10^3$ |
|     Acrobot | $100 \times 10^3$ |
|     MountainCar | $200 \times 10^3$ |
|     LunarLander | $200 \times 10^3$ |
| *Ranges* | |
|   minibatch size | {128, 256, 512} |
|   max gradient norm | [0.25, 1.00] |
|   learning rate | [0.0001, 0.001] |
|   exploration ratio | [0.05, 0.15] |
|   discount factor | [0.980, 0.999] |
|   gradient steps before target network update | [500, 2000] |

## A.5 Examples of Epistemic Uncertainty Behaviour on Individual Runs of Gym Environments

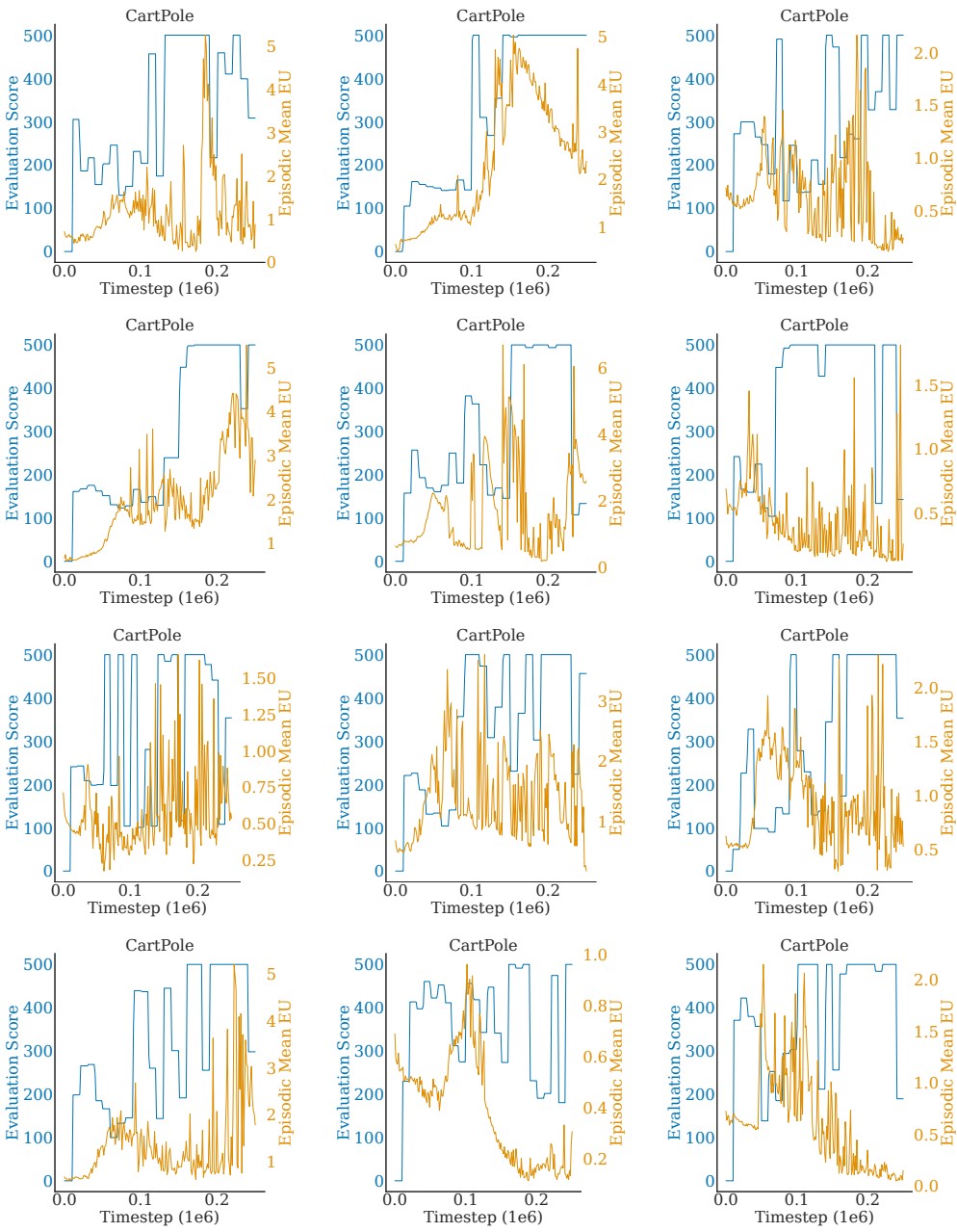

Figure 6: Episodic mean epistemic uncertainty and evaluation performance curves on various runs of Cart-Pole.

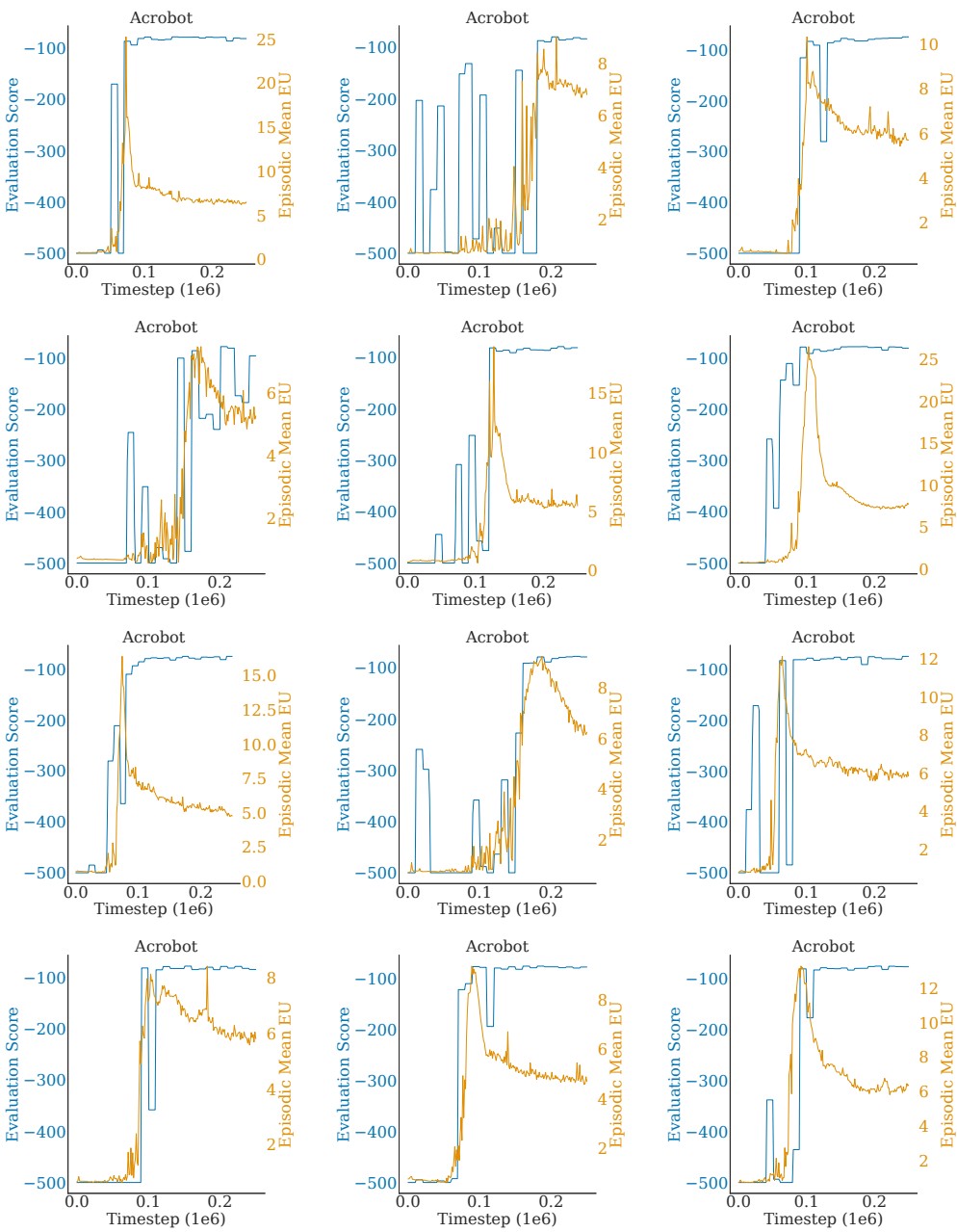

Figure 7: Episodic mean epistemic uncertainty and evaluation performance curves on various runs of Acrobot.

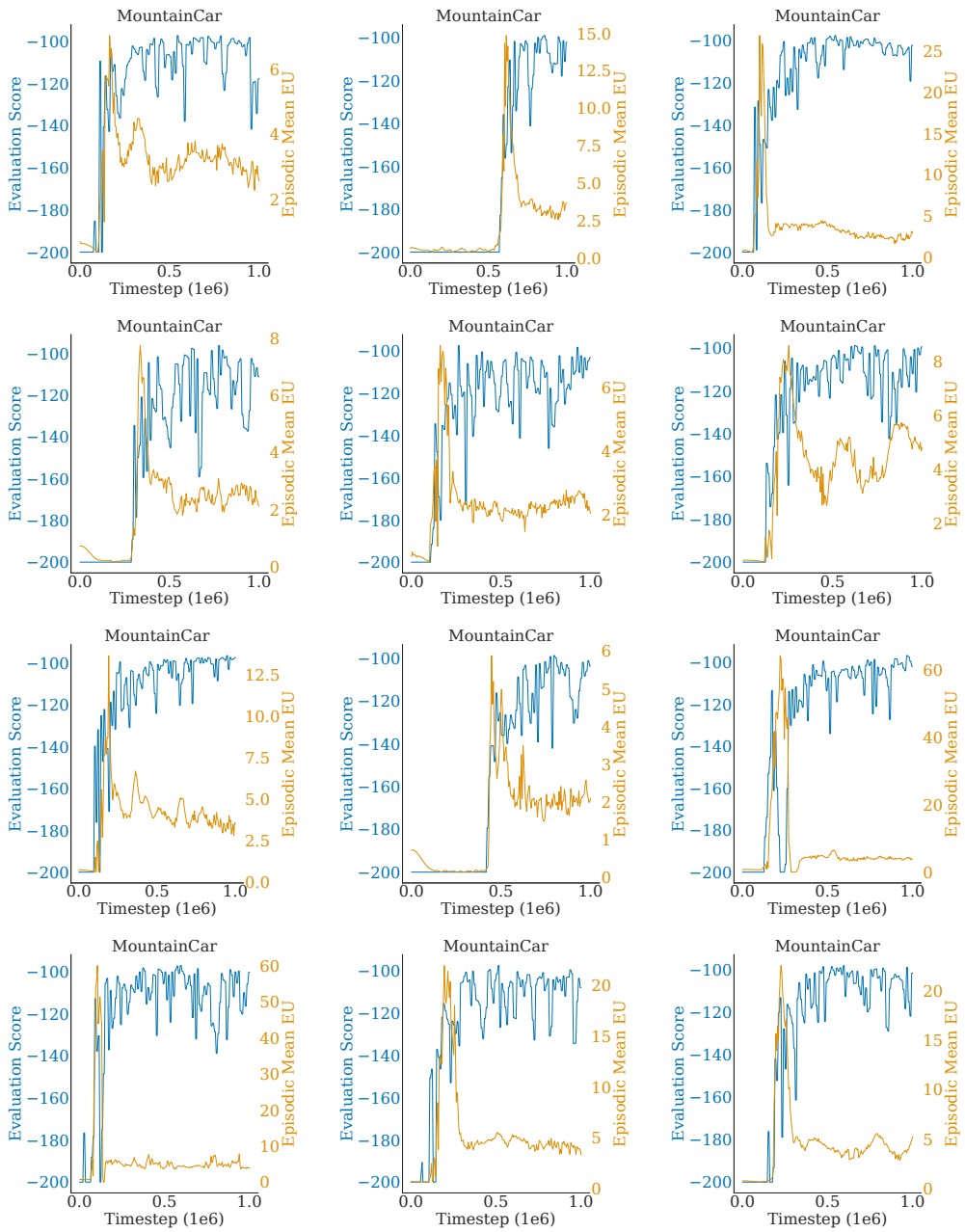

Figure 8: Episodic mean epistemic uncertainty and evaluation performance curves on various runs of MountainCar.

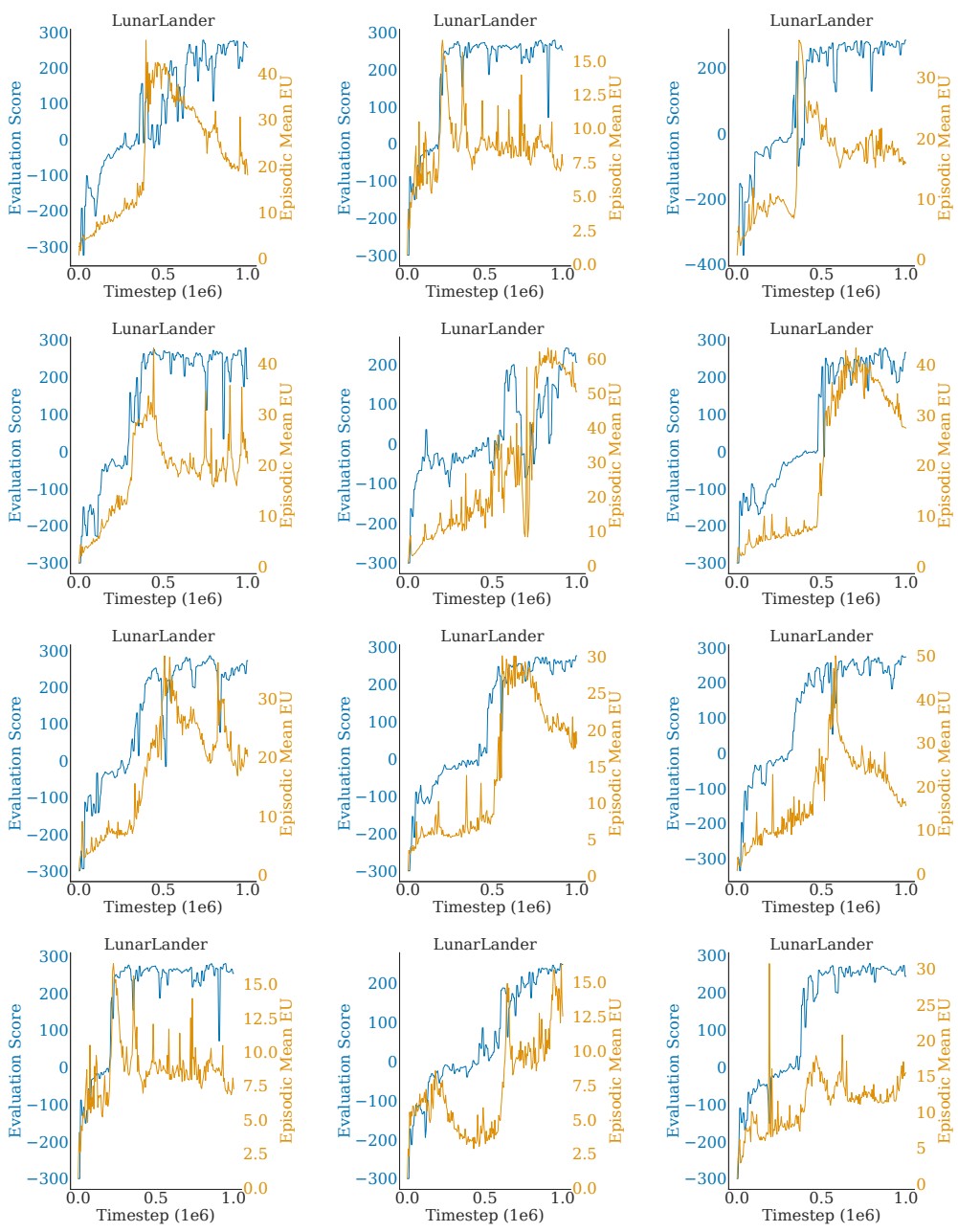

Figure 9: Episodic mean epistemic uncertainty and evaluation performance curves on various runs of LunarLander.

# B   Choosing Confidence Scale

CCGE introduces one hyperparameter — the confidence scale, $\lambda$ — on top of the base RL algorithm. We perform a series of experiments to study the effect that this hyperparameter has on the learning behaviour of the algorithm. To do so, we perform 100 different runs with $\lambda \in [0, 5]$, and compute mean values over 200k timestep intervals. We plot the guidance ratio — the proportion of transitions where the learning policy requests guidance from the oracle policy — as well as the average evaluation performance for the PyFlyt/QuadX-Waypoints-v0 and Walker2d-v4 environments. The results are shown in Fig. 10.

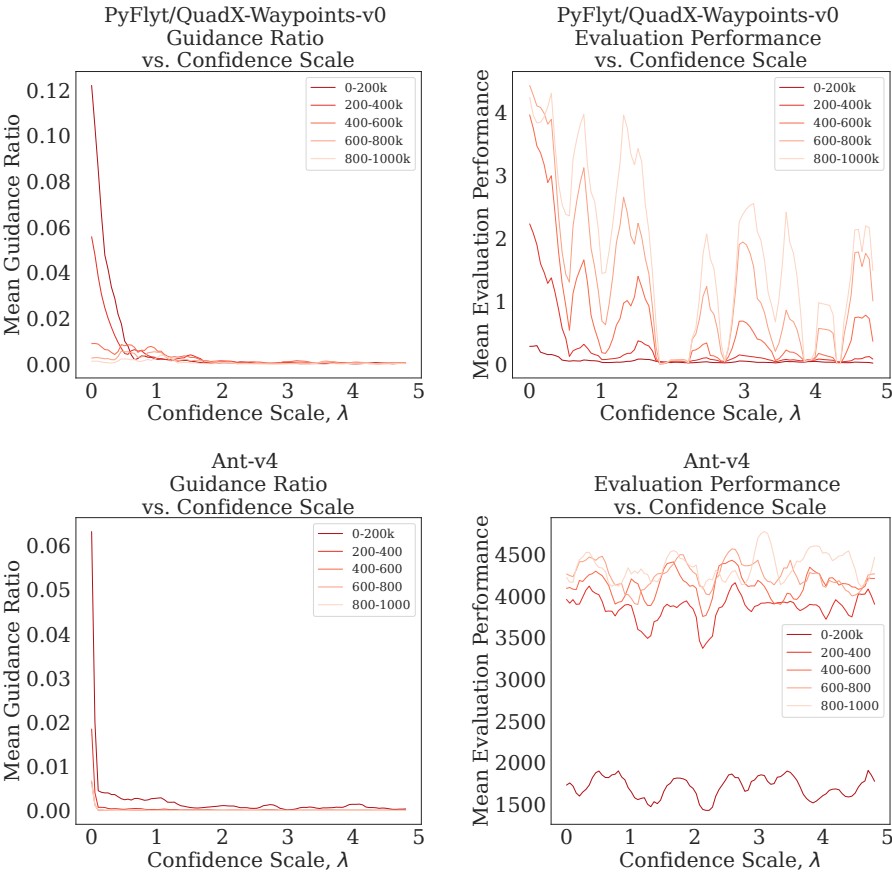

Figure 10: Mean guidance ratio and mean evaluation performance of CCGE on one sparse and one dense reward environment. The results are averaged over 200k timestep intervals, each line corresponds to 100 different runs using a range of values for $\lambda$.

The choice of environment for these sets of experiments, while not exhaustive, were chosen to study the effect of $\lambda$ on CCGE in both a dense and sparse reward environment. As expected, low values of $\lambda$ result in a higher guidance ratios, and this is especially true early on in training (before 400k timesteps). At later stages in training, the effect of $\lambda$ is almost nullified, likely due to the learning policy's performance surpassing that of the oracle policy in all states. In sparse reward environments, a high guidance ratio leads to better evaluation and learning performance; while in dense reward settings, this effect seems to not be present.

As an overarching recommendation, we suggest $\lambda \gtrsim 0$ for more sparse reward settings, higher values seem to have no effect on performance for well-formed dense reward environments.

## C  Exploring Performance Degradation in Ant-v4

The performance of CCGE starts degrading after about 300k timesteps. We suspect that this is not an issue of instability in CCGE, but a result of catastrophic forgetting due to the limited capacity of the FIFO replay buffer. Several additional experiments were performed to pinpoint this reasoning. Here, we train both SAC and CCGE in Ant-v4 for 2 million timesteps, with a replay buffer size of $3 \times 10^5$ and $5 \times 10^5$. In addition, we also implement a global distribution matching replay buffer (Isele & Cosgun, 2018) for both replay buffer sizes for both learning algorithms. The IQM results for each configuration, displayed in Fig. 11, were aggregated using 20 random initial seeds.

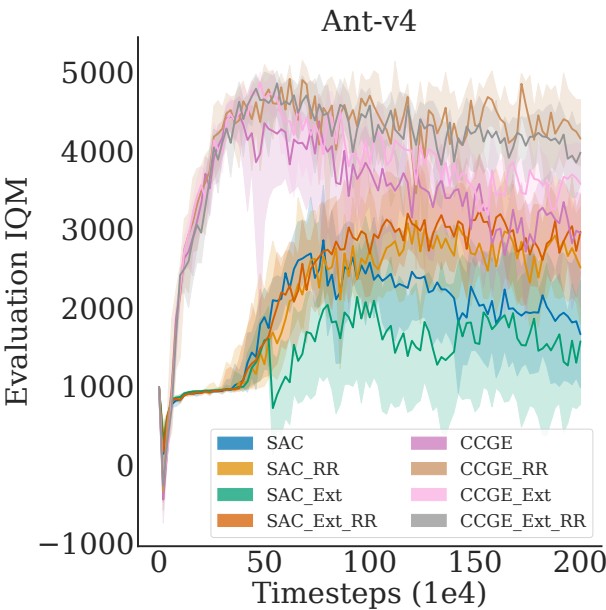

Figure 11: Learning curves of CCGE and SAC on the Ant-v4 environment, using different replay buffer sizes and different replay buffer forgetting techniques, trained for 2 million timesteps. Runs with the `_Ext` extension denote experiments done with a replay buffer size of $5 \times 10^5$, and runs with the `_RR` extension denote experiments where global distribution matching was used as a forgetting technique.

From the results, while the performance of CCGE in the default configuration does degrade to the peak level of SAC, the performance of SAC also ends up degrading a significant amount when training is continued for an extended amount of time. Utilizing a larger replay buffer size does aid in reducing this performance degradation in CCGE, but seems to hurt performance in SAC. When using global distribution matching — a technique for circumventing catastrophic forgetting — the performance degradation of both CCGE and SAC is much less severe, inline with the results obtained by Isele & Cosgun (2018).

The results here are interesting, posing an interesting question for future research on whether CCGE can be used to reduce the replay buffer size through an oracle policy. That said, the results also suggest that the performance degradation is not necessarily instability, nor is it an artifact caused by CCGE's implementation alone as it is also present in SAC.

## D   SAC and CCGE Hyperparameters for Mujoco Tasks

Table 2: SAC and CCGE Hyperparameters for Hopper-v4, Walker2d-v4, HalfCheetah-v4 and Ant-v4

| Parameter | Value |
|---|---|
| *Constants* | |
| optimizer | AdamW (Loshchilov & Hutter, 2018; Kingma & Ba, 2015) |
| learning rate | 4e-4 |
| batch size | 256 |
| number of hidden layers (all networks) | 2 |
| number of neurons per layer (all networks) | 256 |
| non-linearity | *ReLU* |
| number of evaluation episodes | 100 |
| evaluation frequency | every $10 \times 10^3$ environment steps |
| total environment steps | $1 \times 10^6$ |
| replay buffer size | $300 \times 10^3$ |
| target entropy | $-\dim(\mathcal{A})$ |
| discount factor ($\gamma$) | 0.99 |
| *For CCGE only* | |
| confidence scale ($\lambda$) | 1.0 |

## E   AWAC, JSRL, and CCGE Hyperparameters for AdroitHand Tasks

We utilize the same SAC backbone for all three algorithms. For JSRL, we utilize JSRL Random as described in the original paper (Uchendu et al., 2022), supposedly a more performant version of JSRL which allows the oracle policy to act for a random amount of timesteps in each episode before the learning policy takes over.

Table 3: AWAC, JSRL, and CCGE Hyperparameters for AdroitHandDoorSparse-v1, AdroitHandPen-v1 and AdroitHandHammer-v1.

| Parameter | Value |
|---|---|
| *Constants* | |
| optimizer | AdamW (Loshchilov & Hutter, 2018; Kingma & Ba, 2015) |
| learning rate | 4e-4 |
| batch size | 512 |
| number of hidden layers (all networks) | 2 |
| number of neurons per layer (all networks) | 128 |
| non-linearity | *ReLU* |
| number of evaluation episodes | 100 |
| evaluation frequency | every $10 \times 10^3$ environment steps |
| total environment steps | $1 \times 10^6$ |
| replay buffer size | $300 \times 10^3$ |
| target entropy | $-\dim(\mathcal{A})$ |
| discount factor ($\gamma$) | 0.92 |
| *For CCGE only* | |
| confidence scale ($\lambda_{\text{CCGE}}$) | 1.0 |
| *For AWAC only* | |
| Number of demonstration transitions | $100 \times 10^3$ |
| Pretrain epochs | 10 |
| Lagrangian multiplier ($\lambda_{\text{AWAC}}$) | 0.3 |

## F   AWAC, JSRL, and CCGE Hyperparameters for PyFlyt Tasks

Part of the observation space in the PyFlyt Warpaint environments uses the `Sequence` space from Gymnasium. As a result, using a vanilla neural network to represent the actor and critic is insufficient due to the non-constant observation shapes. We utilize the network architectures illustrated in Figure 12, which takes inspiration from graph neural networks to process the waypoint observations and agent state separately. All algorithms utilize the same architecture. For JSRL, we use JSRL Random — the more performant version of JSRL as described in the original paper(Uchendu et al., 2022).

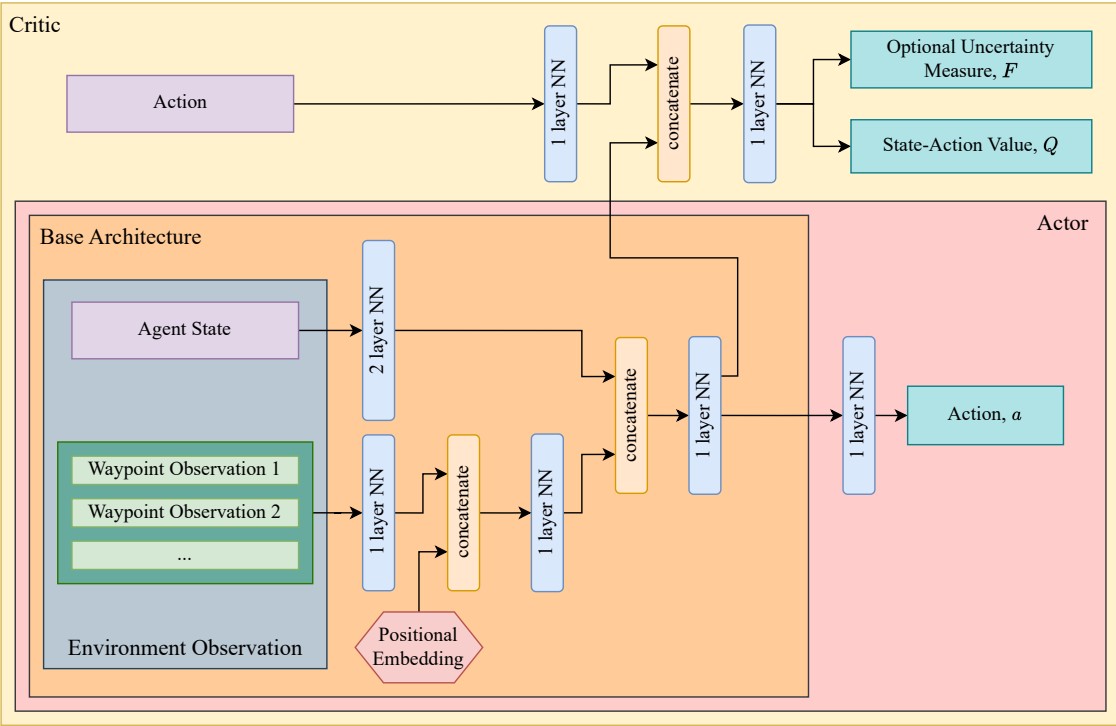

Figure 12: Block diagram of the architectures of the actor and critic used for PyFlyt experiments.

Table 4: AWAC, JSRL, and CCGE Hyperparameters for PyFlyt/Fixedwing-Waypoints-v0 and PyFlyt/QuadX-Waypoints-v0.

| Parameter | Value |
|---|---|
| *Constants* | |
|    optimizer | AdamW (Loshchilov & Hutter, 2018; Kingma & Ba, 2015) |
|    learning rate | 4e-4 |
|    batch size | 1024 |
|    number of neurons per layer (all networks) | 128 |
|    non-linearity | *ReLU* |
|    number of evaluation episodes | 100 |
|    evaluation frequency | every $10 \times 10^3$ environment steps |
|    total environment steps | $1 \times 10^6$ |
|    replay buffer size | $300 \times 10^3$ |
|    target entropy | $-\dim(\mathcal{A})$ |
|    discount factor ($\gamma$) | 0.99 |
| *For CCGE only* | |
|    confidence scale ($\lambda_{\mathrm{CCGE}}$) | 0.1 |
| *For AWAC only* | |
|    Number of demonstration transitions | $100 \times 10^3$ |
|    Pretrain epochs | 10 |
|    Lagrangian multiplier ($\lambda_{\mathrm{AWAC}}$) | 0.3 |

