# OpenReview forum: "Some Supervision Required: Incorporating Oracle Policies in Reinforcement Learning via Epistemic Uncertainty Metrics"
_TMLR — Rejected by TMLR_

### Review · Reviewer_ytg1 · 2023-05-20

**Summary Of Contributions:**

The paper proposes Critic Confidence Guided Exploration (CCGE) an empirical approach to incorporate the experience collected with an oracle policy into actor-critic architectures. The approach is based on estimating the "potential improvement" of an action and, based on that value, deciding whether to follow the oracle (both in training the critic and for the played policy) or to trust the current value of the critic. This requires estimating the epistemic uncertainty of the critic, for which the authors provide two approaches: implicit and explicit. The resulting algorithm is tested on Mujoco environments and on some hard-exploration tasks.

**Audience:**

No

**Broader Impact Concerns:**

None.

**Claims And Evidence:**

No

**Requested Changes:**

* Please provide a more detailed discussion about the hyperparameter $\lambda$ and a sensitivity analysis.

* Please motivate the non-fully satisfactory results in the Mujoco tasks.

**Strengths And Weaknesses:**

***Strengths***
* The paper addresses an important problem, i.e., how to incorporate the possible experience coming from an oracle in a principled way to avoid compromising the learning process.
- Some of the experimental results are able to show some advantages of the proposed approach over the baselines.

***Weaknesses***
* [Hyperparameter] The decision on whether to trust the critic or follow the oracle suggestion is based on a threshold $\lambda$ called the "confidence scale". This seems to be a crucial hyperparameter that, if wrongly chosen can compromise the final results and, consequently, it seems to be a limitation of the approach. The authors provide a suggestion on values of $\lambda$ depending on whether the task requires a hard exploration. I am wondering if this parameter can be selected in a theoretically-principled way? Have the authors conducted any sensitivity analysis on it?

* [Experimental results] For what concerns the Mujoco tasks, the experimental results do not provide a clear advantage overall. Indeed, for the first three environments the advantage is very small, while for the Ant, we see the ability to reach higher performances (in the case of the best oracle) but the performance, then, experiences a reduction. Is this an alert of instability? I wonder, in the case the training was conducted for other steps if the performance would have reached that of SAC.

***Minor***
* The preliminaries section misses the definition of Q-function and, overall, should be more detailed on the definition of MDP, value function and optimality.
* Avoid abbreviations like "doesn't" -> "does not"

---

> ### Author Response · Authors · 2023-06-16
> **Response to Reviewer YTG1**
>
> We thank the reviewer for their comments and responses, the feedback has been very useful and we think the additional experiments we add will bring more value to the work at hand.
>
> # Regarding Weaknesses
> **Hyperparameter** We agree that this hyperparameter (confidence scale, $\lambda$) plays a fairly crucial role in the performance of CCGE, especially in sparse reward cases. In essence, the lower $\lambda$ is, the more the learning policy requests guidance from the oracle; on the opposite end, high $
> \lambda$s cause the learning algorithm to regress into the base reinforcement learning algorithm instead (SAC in our case).
>
> In response to this, we have performed additional experiments where we vary $\lambda$ from 0.0 to 5.0 with 100 random seeds in order to study the behaviour of CCGE with respect to amount of guidance requested and evaluation performance.
> This has been done for one dense and one sparse reward environment.
>
> In this case, we observed that there is no performance degradation to having $\lambda$ be too low, a value of 0.0 seems to work the best for both cases.
> In addition, we also see that for dense reward environments, the learning policy tends to request a lot less guidance than in a sparse reward environment.
> At present, we do not know why this may be the case, but it could be the reason that the performance on the MuJoCo tasks are less than satisfactory.
>
> **Experimental Results** Ant's performance degrades over time as a consequence of the replay buffer forgetting.
> In general, for limited capacity FIFO buffers, performance of off-policy RL algorithms degrades severely once state coverage for suboptimal actions is no longer present.
> We point the reviewer towards the following manuscript studying this behaviour in more detail [1].
>
> In the case of CCGE in Ant, we see that the performance drop happens after 300k - the point where the replay buffer starts forgetting the oldest experiences.
> To explore this more concisely, we have performed experiments in the Ant environment, where we train agents for 2M transitions (2x those previously done in our work).
> Here, we train SAC and CCGE with replay buffer sizes of 3e5 and 5e5, with a FIFO buffer as well as with "global distribution matching".
> We observed that, yes, CCGE's performance does eventually degrade to the level of SAC under the default settings, but SAC's performance actually degrades to be close to the performance of a randomly initialized policy.
>
> The results here are interesting, and would pose an interesting future research question on whether CCGE can be used to reduce the replay buffer size through an oracle policy.
> That said, the results also suggest that the performance degration is not necessarily instability, nor is it an artifact caused by CCGE's implementation alone as it is also present in SAC.
>
> # Changes
> 1. We added a sensitivity analysis for the $\lambda$ parameter in Appendix B.
> 2. We addressed the issue of performance degradation in Ant-v4 in Appendix C.
>
> [1] Selective Experience Replay for Lifelong Learning, AAAI 2018

---

> > ### Comment · Reviewer_ytg1 · 2023-08-11
> > **Re: Response to Reviewer YTG1**
> >
> > I thank the authors for the feedback and for having added the listed experimental results.

---

### Review · Reviewer_JJKE · 2023-06-14

**Summary Of Contributions:**

The authors introduce a novel exploration method for deep RL which leverages an oracle policy. The oracle is used in two ways 1) to directly take actions in the environment and 2) to provide a supervised loss (e.g., minimize the MSE between the learning policy’s actions and the oracle’s actions in continuous action spaces) on behavior. These two conditions are invoked when the agent’s upper confidence bound on the value of the oracle action is sufficiently greater than that of its own action, the intuition being that it follows the oracle when it has more epistemic uncertainty. Epistemic uncertainty is estimated in two possible ways, either 1) “implicitly” as the variance of Q-function estimates or 2) “explicitly” as the square root of the discounted sum of Bellman errors. The approach is evaluated on a set of continuous control and sparse-reward robotics tasks.


**Audience:**

Yes

**Broader Impact Concerns:**

I don't have any broader impact concerns regarding this work.

**Claims And Evidence:**

No

**Requested Changes:**

- Adding more ablations and baselines (see above) would be most important for me, both to establish the components of CCGE which are important for its performance and to ensure a fair comparison.

- Secondarily, though still important, are results which foster understanding of the method (see third bullet point above under Weaknesses).

- Additional plots showing sample complexity which take into account the samples required to train the oracle (not critical, but I think at minimum a discussion/acknowledgment of this is important).

**Strengths And Weaknesses:**

*Strengths*

- The major strengths of the approach are its simplicity and low computational cost.

- The paper is also clearly written and mostly easy to follow.

- Empirically, the method appears to perform well (though see below for further discussion) and the high number of random seeds and use of IQM for evaluation indicate that the results are reliable.

*Weaknesses*

- In my view, the primary weakness of the paper is a lack of ablations / comparison to other baselines. Specifically, why not compare against using an upper confidence bound on the Q-function with SAC? Optimistic actor-critic [1] seems like a natural baseline approach which captures this idea. There is a significant literature on policy optimization approaches with optimism / incorporating uncertainty which don’t require an oracle policy and to which no comparison is provided. Other ablations would be to measure the performance of CCGE only using the oracle to take actions when uncertain or only using the oracle to provide a supervisory loss, rather than both. What about behavior-cloning the oracle policy and then fine-tuning with an exploration bonus / optimistic Q-functions to escape local minima?

- Obtaining the oracle policy in the robotics and PyFlyt domains by training SAC in the dense reward version of the environment seems a bit unfair, as it gives the overall CCGE approach a leg up on the other methods (at the minimum, I think the # of samples used to train the oracle policy should be counted on the x-axis / towards the overall complexity of the algorithm—this applies to the Mujoco environments as well). In the case of applications to truly sparse domains, it doesn’t seem like it would be possible to obtain an oracle policy in this way.

- Beyond the performance plots, there aren’t any results which give a deeper understanding of how the algorithm is working—e.g., how often does the agent query the oracle over time during training?

- There are no formal guarantees for / performance analysis of the proposed method.

- Minor: 1) - The characterization of the SAC policy objective (Eq. 2) is strange—omitting the $\log \pi$ term is omitting one of the key components of the approach. Why not just write the actual objective? 2) - The description of the SAC critic as a modified DQN is incorrect. DQN is based on Q-learning, while the Q-functions used in SAC are just for (soft) policy evaluation, not control.

[1] https://arxiv.org/abs/1910.12807

---

> ### Author Response · Authors · 2023-07-13
> **Response to Reviewer JJKE**
>
> We thank the reviewer for their insights and comments, we have added various changes that we think add more value to the work.
>
> # Regarding Weaknesses
> **More Ablations**
> There seems to be some confusion on the purpose of CCGE.
> CCGE is not intended as a new exploration method, but rather a technique of leveraging the experience of an oracle policy into actor-critic architectures instead of learning from scratch.
> Perhaps this confusion is due to the term "exploration" in the name of the algorithm.
> For that reason, we did not compare CCGE against other baselines that utilize uncertainty to promote exploration.
> We agree that it is possible to compare CCGE with OAC, but the option between CCGE and OAC is not an either/or decision - one can utilize CCGE whenever the the critic thinks that the oracle's actions are of higher value, and explore optimistically using OAC otherwise, effectively incorporating both ideas into the same algorithm.
> This idea is indeed interesting, and while we would be happy to pursue results in this direction, we think that experiments of this sort are out of the scope of the paper itself.
> We do agree that it would be interesting to observe the effect of either supervision only or guidance only against the full CCGE algorithm, and we have indeed added a set of experiments into the main text that covers this aspect.
> In addition, while it is possible to pretrain a policy on oracle actions and then perform optimistic finetuning, previous works [1] that have explored this idea have shown that this leads to a poorly initialized critic, leading to immediate performance collapse that never recovers.
>
> **More Understanding**
> In addition to the experiments for determining whether supervision or guidance matters most, we have further added a set of experiments which study the proportion of agent queries over the duration of training against different confidence scales.
> We hope that this provides some insight into how often the learning policy is requesting guidance from the oracle policy.
>
> **Oracle Sample Complexity**
> CCGE's core assumption is that an oracle policy is already available, and the problem statement is on how to leverage its experience in a matter that does not compromise the learning process.
> If an oracle policy is not present, then we agree that it is much better to perform exploration via other means instead of training an oracle policy just to utilize CCGE.
> Under this problem statement, we do not think that the sample complexity required to train the oracle policy matters since this oracle can be obtained via other methods, such as hard coding it.
> That said, we have added in a statement of the number of samples taken to train the oracle policies into the main text.
>
> # Changes
> 1. We have added a comparison of CCGE using guidance, supervision, or both in Section 5.1.
> 2. A study of how much the learning policy requests guidance is also shown in Appendix B.
> 3. We have added statements within Section 5 indicating the number of environment steps in SAC required to train the oracle policies.
>
> [1] Figure 2 in Jump-Start Reinforcement Learning, PMLR 2023

---

> > ### Comment · Reviewer_JJKE · 2023-08-05
> > **Response**
> >
> > I'd like to thank the authors for taking the time to respond to my comments and modify the paper. Please find my responses below:
> >
> > **Ablations** It seems strange to me to say that Critic Confidence Guided Exploration is not an exploration method, especially when statements like these are in the paper: "We term this algorithm CCGE, in reference to the learning policy mimicking the oracle policy for exploration." In particular, given the results in Figure 3, "Guidance Only" CCGE seems to essentially match the performance of Supervision + Guidance across domains, which makes it seem as though the performance gain must mainly originate in having access to what is in essence a good exploration policy early in training. I do accept that the way OAC does exploration is largely orthogonal to CCGE and so a comparison isn't necessary (but would be a bonus). Thank you for the pointer to [1], that is an interesting result.
> >
> > **More Understanding** Thank you for adding these plots! They indeed seem to show that the agent relies more on the oracle early in training, as would be expected.
> >
> > **Oracle Sample Complexity** Thank you for your response, I believe this is a valid point in cases where the oracle has indeed been given to the agent externally. However, in the current set-up, the oracle is simply another policy pre-trained for the purpose of training the current policy. It might have been better / more convincing to have at least one truly "already available" oracle policy. Otherwise, it stands to reason that one could imagine a chain of oracles each guided by the previous one.
> >
> > Thank you once again! This is an interesting work.

---

### Review · Reviewer_benJ · 2023-07-10

**Summary Of Contributions:**

The paper focuses on learning to control in the presence of oracle policies, and how such oracle policies can be used to guide an RL agent's learning. The proposed method simply chooses the oracle policies' actions when the RL agent is uncertain, where uncertainty is w.r.t. the Q function of the agent. The paper shows that this is a fairly effective way to incorporate expert knowledge while continuing to train the policy.

**Audience:**

Yes

**Claims And Evidence:**

Yes

**Requested Changes:**

I would appreciate it if the authors can provide a few results on the last point above, i.e. showing that choosing actions from the oracle policy at the right learning step actually leads to faster decrease in uncertainty. This would certainly boost my understanding of the method.

**Strengths And Weaknesses:**

Strengths:
- The paper is well written and the experiments are well structured. The related work discussion is quite detailed and the proposed method is simple.

Weakness:
- The performance improvements seem minimal and inconsistent compared to baseline methods.
- More importantly, the biggest weakness in my opinion is that the authors never show that their method is able to capture a good estimate of uncertainty and then make the right choice when deciding between the agent's action vs the oracle policy's action. The uncertainty results shown in the Appendix are for a DQN agent and so do not shed light on whether the idea of choosing between the two candidate actions based on the uncertainity is actually beneficial.

---

> ### Author Response · Authors · 2023-07-13
> **Response to Reviewer benJ**
>
> We thank the reviewer for the comment.
>
> We think that there is some confusion for the purpose of CCGE.
> CCGE's main goal is not to reduce uncertainty as fast as possible, it is instead to leverage the experience in the oracle policy to speed up agent learning, using uncertainty as a proxy to do so.
> The experiments shown in the Appendix A are instead meant to show that our proposed "explicit epistemic uncertainty estimation" technique serves to provide a reasonable method for estimating the uncertainty in the critic, and not that it decreases with more samples during training.
> Validating whether this method actually provides a good estimate is difficult beyond simple visual checks of learning curves.
>
> That being said, our suggested "implicit epistemic uncertainty estimation" is a well known method for approximating the epistemic uncertainty within Q-value functions, and has been used in various past works [1][2][3], so there should be little doubt that this provides a reasonable estimate of epistemic uncertainty - especially as the number of networks increases.
>
> We have run CCGE using both techniques for measuring uncertainty in Figure 1, where the learning curves show that CCGE's performance is almost equal with both techniques.
> This suggests that, since the "implicit epistemic uncertainty estimation" technique is a reasonable measure, the "explicit epistemic uncertainty estimation" technique should also be reasonable, given that they both perform similarly in the context of CCGE.
> As to whether this is the best technique for leveraging uncertainty is hard to say.
>
> Perhaps the reviewer may find interest in how the learning policy automatically weans off the oracle policy as training progresses, leading to better upperbound Q-value estimates for both its action and the oracle's via better estimation of both Q-values and estimated epistemic uncertainty.
> We have conducted some experiments in this regard, and the results are shown in Appendix B.
>
> We hope this clears things up.
>
> [1] Better Exploration with Optimistic Actor-Critic, NeuRIPS 2019
>
> [2] Estimating Risk and Uncertainty in Deep Reinforcement Learning, ICML 2020
>
> [3] Deep Exploration via Bootstrapped DQN, NeuRIPS 2016

---

### Decision · Action_Editors · 2023-08-30

**Recommendation:** Reject

**Comment:**

The main concern of the reviewers is whether the claims are sufficiently supported, and there was unanimous agreement that it is too weak in this area in its current form.  See the specific notes above.

**Audience:**

The paper is studying a problem of particular interest to TMLR's audience.  Although there's some confusion in not just exposition but in experimental evidence as to what the role the proposed algorithm is being claimed to address.

**Claims And Evidence:**

There's some confusion in not just exposition but in experimental evidence as to what the role the proposed algorithm is being claimed to address.  Is it about a better exploration method that uses an expert policy?  Or is it about how to handle epistemic uncertainty through using an expert policy?  Or is it about how to use an expert policy in a safe way?

Each of these would lend to a different set of claims and experiments that would support these claims.  The role of the algorithm needs to be more clear.  What claims then are being made based on that role?  And are there the right experiments/ablations to support those claims?  For example, even if the role of the algorithm is not about exploration, the fact is that some of how the algorithm being used is in giving optimism when the agent is uncertain.  Teasing apart whether the benefits observed is due to the reasons claimed is critical.

Lastly, there's some concerns that the degree of improvement in some cases is quite small, further adding to the weakness in supporting claims.  I think a more clear set of claims and supporting evidence though is more important than the degree of observed improvement.

**Resubmission Of Major Revision:**

The authors may consider submitting a major revision at a later time.